# Enhancer RNAs (eRNAs) in Cancer: The Jacks of All Trades

**DOI:** 10.3390/cancers14081978

**Published:** 2022-04-14

**Authors:** Sara Napoli, Nicolas Munz, Francesca Guidetti, Francesco Bertoni

**Affiliations:** 1Institute of Oncology Research, Faculty of Biomedical Sciences, USI, 6500 Bellinzona, Switzerland; nicolas.munz@ior.usi.ch (N.M.); francesca.guidetti@ior.usi.ch (F.G.); francesco.bertoni@ior.usi.ch (F.B.); 2Oncology Institute of Southern Switzerland, Ente Ospedaliero Cantonale, 6500 Bellinzona, Switzerland

**Keywords:** enhancer, ncRNA, eRNA, transcriptional regulation, cancer, immune response

## Abstract

**Simple Summary:**

This review focuses on eRNAs and the several mechanisms by which they can regulate gene expression. In particular we describe here the most recent examples of eRNAs dysregulated in cancer or involved in the immune escape of tumor cells.

**Abstract:**

Enhancer RNAs (eRNAs) are non-coding RNAs (ncRNAs) transcribed in enhancer regions. They play an important role in transcriptional regulation, mainly during cellular differentiation. eRNAs are tightly tissue- and cell-type specific and are induced by specific stimuli, activating promoters of target genes in turn. eRNAs usually have a very short half-life but in some cases, once activated, they can be stably expressed and acquire additional functions. Due to their critical role, eRNAs are often dysregulated in cancer and growing number of interactions with chromatin modifiers, transcription factors, and splicing machinery have been described. Enhancer activation and eRNA transcription have particular relevance also in inflammatory response, placing the eRNAs at the interplay between cancer and immune cells. Here, we summarize all the possible molecular mechanisms recently reported in association with eRNAs activity.

## 1. Introduction

Huge progress made in understanding the human genome has come hand in hand with the discovery of its complexity. Only 1–2% of the human transcriptome codes for proteins [1]. The remaining transcriptome is represented by an overwhelming amount of non-coding RNAs (ncRNAs): ribosomal-RNA (rRNA), transfer-RNA (tRNA), small nucleolar RNA (snoRNA), micro-RNA (miRNA), but also long-noncoding-RNA (lncRNA), including competing-endogenous-RNA (ceRNA), and many more [1]. Furthermore, the genome comprises extended regulatory regions such as enhancers, defined as *cis*-regulatory elements spatially and temporally cooperating with promoters, able to bind transcription factors, co-regulators, and RNA polymerase II (RNApol II) to mediate target gene transcription [2]. The study of the formation of the enhancer–promoter loop crucial for the transcription initiation process showed that enhancers exert their function in an orientation independent manner. In addition, enhancers are positioned at various distances upstream or downstream the transcription starting side (TSS) of the associated gene, and they can work together in networks comprising other enhancers to control gene expression [3]. Currently, the knowledge of such complex enhancer interplay is limited mainly to enhancers organized in super enhancer (SE) regions. SEs are large clusters of enhancers (>20 kb in size), forming specific 3D structures associated with massive transcription rates and play an essential role in cell growth and differentiation, but also in disease initiation and progression of cancers [4]. 

The epigenetic state of enhancers is essential for their function and, indeed, it is often exploited to identify enhancers themselves and to define their genomic localization. Nucleosomes surrounding active enhancers harbor high histone H3 lysine 4 mono-methylation (H3K4me1) and histone H3 lysine 27 acetylation (H3K27ac) levels [5]. However, additional factors need to be considered for robust enhancer identification. The discovery of the binding of RNApol II to active enhancers revealed the transcription of a new subclass of ncRNAs, the enhancer RNAs (eRNAs). The latter can be used as an additional indicator for active enhancer regions beside the classical histone marks previously described [6]. Moreover, the discovery of eRNA expression added another layer of complexity to the human transcriptome, leading to intense research on the features and potential functions of this new class of ncRNAs [7]. 

Although more and more commonly identified and recognized, the biological roles of eRNAs are still under investigation. In the last couple of years, a growing number of studies showed that eRNAs participate in the regulation process of gene transcription [7]. First, they contribute to the enhancer–promoter loop, recruiting cohesin and mediator proteins [8,9]. They trap transcription factors, as YY1, to increase their local concentration at DNA at the site of transcription [10]. Furthermore, eRNAs are also directly involved in transcription, acting as decoys for negative elongation factor (NELF) and releasing paused RNApol II to promote the elongation process [11] (Figure 1).

Notably, eRNAs seem to play a role not only in homeostasis and cell development, but also in disease development and differentiation-state. They can modulate the expression of oncogenes, tumor suppressor genes, and inflammatory or cancer signaling pathways by modifying gene transcription and protein–RNA interactions [12]. The eRNAs also represent potential biomarkers of treatment response [12,13,14].

Based on the rapid progress in the characterization of ncRNAs, in this manuscript, we aim to review the latest findings and provide new insights into the features and mechanisms of action of eRNAs, highlighting the huge potential of exploring eRNA interactions in different cancer types.

## 2. eRNAs Instability and Methodology of Detection

eRNAs are noncoding molecules bidirectionally transcribed from enhancers by RNApol II [15]. The eRNAs are 5′-capped but less stable and shorter than mRNAs because they are retained in the nucleus and exposed to exosome-mediated decay [16]. Indeed, they are mainly unspliced and not polyadenylated. The lack of polyA causes the recruitment of exosome, because the small distance between polyA signal and transcription start site prevents the assembly of polyadenylation machinery at RNApol II C-terminal domain (CTD) [17]. eRNAs are densely located on the chromatin where they help the stabilization of enhancer–promoter (E–P) interactions [17]. These features make them rarely detectable by conventional polyA RNA-seq. Their identification and quantification is feasible by techniques such as ribosomal RNA (rRNA)-depleted total RNA-seq or cap analysis of gene expression (CAGE) [18,19,20]. Other techniques such as the global run-on sequencing (GRO-seq) [21] and its variant precision nuclear run-on sequencing (PRO-seq) [22] investigate RNApol II activity, sequencing the nascent RNA molecules, and thus, they allow the detection of even unstable transcripts. Transient transcriptome sequencing (TT-seq) is often applied to study eRNAs [23,24,25]. It is a sensitive approach to detect short-lived transcripts, which combines a short 4-thiouridine (4sU) RNA labeling pulse with an RNA fragmentation step to enrich for newly synthesized RNA. An adapted CAGE protocol, NET-CAGE, was recently developed to study TSSs of native elongating transcripts (NETs) in a strand-specific manner [26]. It sensitively identifies and quantifies true transcriptional activities of enhancers at high nucleotide resolution and allows the study of RNA synthesis and degradation rates at the TSS level. 

Other techniques can help in characterizing eRNA functions, linking the eRNA with its target gene. For instance, an advanced imaging protocol for single molecule fluorescence in situ hybridization (smFISH) was implemented to detect short and lowly expressed transcripts at single cell level [27]. This method uses single biotinylated short (20 nt each) probes, which, post hybridization, allow the binding of multiple copies of fluorophore per probe. The system can be multiplexed to achieve simultaneous detection of both eRNA and the induced nascent gene transcript in the same nucleus to prove their co-localization on the chromatin.

Few technologies are emerging to agnostically infer eRNA function by assessing genome-wide RNA chromatin interactions. RADICL-Seq captures proximal RNA-chromatin interactions in a genome-wide manner [28]. Thanks to the enhancer–promoter looping, this technique can capture the spatial proximity of the nascent RNA with the enhancer region. This improved method overcomes limitations of similar pre-existing techniques such as mapping RNA–genome interactions (MARGI) [29], chromatin-associated RNA-sequencing (CHAR-seq) [30], and global RNA interaction with DNA by deep-sequencing (GRID-seq) [31].

## 3. eRNAs in Development and Differentiation

The expression of eRNAs is developmental stage and cell-type specific [32,33]. eRNAs transcribed during cellular differentiation are correlated to their target gene expression [2,34] and their production explains the interaction between enhancers and transcription initiation and their proximity to promoters. Enhancer transcription is the most common rapid transcriptional adjustment occurring when cells undergo the first steps toward a state change [19]. Once the target promoter is activated, enhancer activity is no longer required, and so, after a rapid burst of production and activity, eRNAs frequently return to baseline expression levels. In some specific cases, enhancers are rapidly activated and then continuously expressed and these eRNAs might have additional functional roles still unknown [19]. 

Accordingly, with their pivotal function, eRNAs are involved in determining the developmental stage of cells, often cooperating to regulate the expression of their target genes [2]. General insights into the order of events that establish active enhancers have been recently provided by the application of a genome-wide multi-omics approach to study the C/EBPα-induced trans-differentiation of human precursor leukemia B cells to macrophage-like cells [25]. In particular, TT-seq allowed the identification of transcriptionally active enhancers, then paired with their putative target promoters, and the analysis of changes in transcription activity of enhancers and promoters over time [25]. Most enhancers drive the expression of a common target gene acting in an additive manner, but some enhancers can cooperate synergistically at specific loci to drive target gene transcription [25]. 

The interacting network of molecules that regulates differentiation is complex and involves different types of lncRNAs. A genome-wide study combined RNA sequencing (RNA-seq) and RNA reverse transcription-associated capture sequencing (RAT-seq) to map interacting targets for lncRNAs [35]. The lcnRNA Peblr20 was identified as a demanding element that helps in maintaining the pluripotent status of induced pluripotent stem cells. Peblr20 knockdown in induced pluripotent stem cells withdrew pluripotency state and, vice versa, overexpression of Peblr20 activated stemness-like genes such as Pou5F1, SOX2, and NANOG in fibroblasts, leading to improved pluripotent reprogramming. Notably, lncRNA Peblr20 recruited TET2 to the Pou5F1 enhancer site, leading to DNA demethylation at the Pou5F1 enhancer and, therefore, to Pou5F1 eRNA expression. Pou5F1 eRNA expression itself seemed to modulate the enhancer promoter looping, activating Pou5F1 expression and induced pluripotent state. Thus, Peblr20 utilizes a novel trans epigenetic eRNA activation mechanism to control the stem cell fate mediated by another ncRNA subgroup, suggesting a widely unexplored interplay between other noncoding regulatory RNAs and eRNAs [35]. 

## 4. eRNAs Control Transcription by Several Layers of Sequence Specific Mechanisms

Recently, the abundance of m6A on eRNAs in correspondence of RRACH motif was demonstrated, suggesting an m6A deposition similar to that on mRNAs [36]. Lee at al. showed that the m6A reader YTHDC1 catalyzes the binding of BRD4 to m6A eRNA. In a positive feedback loop, YTHDC1 directly activates enhancers and induces eRNA transcription [37]. However, no evidence of a relationship between m6A and eRNA stability is proven, but on the other hand m6A eRNA bound by YTHDC1 helps chromatin condensation, cross talking to other coactivators and impacting enhancer activity [37]. The act of enhancer transcription generates eRNAs, and the length of these transcripts may influence their m6A levels, consequently facilitating transcriptional activation [37].

Enhancers can be active, primed, or poised, depending on different epigenetic states: active enhancers, which are H3K27ac and H3K4me1 positive and bound by the histone acetyltransferase P300, are strong transcription activators, while, when primed, they have lower H3K27ac and only drive basal transcriptional activation. On the contrary, poised enhancers present H3K4me1 and P300, but they also have the repressive histone mark H3K27me3, associated with Polycomb Repressive Complex 2 (PRC2) silencing [38]. Expression of eRNAs can be activated in poised enhancers by sequence specific mechanism driven by lncRNAs, forming triple-helix structures with DNA [39]. Triplex-based recruitment of chromatin-modifying complexes may represent a common targeting mechanism for enhancer activation. Blank-Giwojna et al. demonstrated that the antisense RNA KHPS1 forms a RNA–DNA triplex at the SPHK1 enhancer, with consequent recruitment of E2F1 and p300 [39]. The enhancer activation induces transcription of an eRNA that is required for SPHK1 expression and cell proliferation. Genomic deletion of the triplex-forming region (TFR) or prevention of KHPS1 binding to DNA impairs cell proliferation and viability [39]. The functional role of the sequence specific lncRNA–DNA binding is proven by the fact that changes in the lncRNA sequence, driving it to distinct genomic location, activating different enhancers and the expression of different genes [39]. The mechanism that leads to promoter activation involves the eRNA-mediated eviction of CTCF, which insulates eSPHK1 from the SPHK1-C promoter [39]. Numerous studies have shown that CTCF regulates enhancer–promoter interactions and the human genome contains thousands of CTCF binding sites [40,41]. Transcription of eRNA may represent a common mechanism allowing neighboring genes to be differentially regulated. A recent paper from Oh et al. showed that regulatory enhancers have the potential to engage more than one promoter in a hosting domain [42]. Different enhancer features, including the level of CTCF, can determine the preference of a specific enhancer in activating different promoters. They referred to this process as “enhancer release and retargeting, ERR” [42]. Importantly, this redirection mechanism of the transcriptional process is at the basis of activation of disease-causing genes [42].

## 5. Noncoding Mutations and Enhancer Hijacking

Germline and somatic single nucleotide variants have been described in enhancers and some of them have been associated with carcinogenesis, inflammatory disorders, cardiomyopathy, and neurodegeneration [32,33,43,44]. Although a portion of them is responsible for alteration of DNA binding by transcription factors [43], this observation enforces the idea that pathogenic mechanisms could involve eRNAs deriving from mutated enhancers [15,33,45].

Among the most common somatic genetic aberrations in cancer, copy number alterations (CNAs) lead to changes in gene dosage of the transcriptional units spanning the altered regions [46]. Chromosomal rearrangements are another common mechanism leading to deregulation of cancer genes, and a possible mechanism is the relocation of regulatory DNA elements including chromatin regions populated with active enhancers, a process dubbed as ‘enhancer hijacking’ [47]. 

Although MYC overexpression has long been known to occur as result of t(8;14) MYC/IgH gene rearrangement [48], most recently, Zhang et al. showed that duplication of lineage-specific 3′ super-enhancers region to MYC (MYC-LASE) increased MYC expression [49]. Decreased expression of MYC has been observed by CRISPR/KRAB–dCas9 mediated repression or deletion of a constituent enhancer within the MYC-LASE region [49]. Similarly, amplifications of super-enhancers on chromosome segment 13q22.1 are found in different cancer types and activate KLF5 oncogene expression in squamous cell carcinomas [50]. In particular, CRISPR/KRAB–dCas9 mediated multiplexed repression in BICR31 cells of specific enhancers revealed they exert a combinatorial effect on KLF5 activation [50]. In leukemia, intrachromosomal inversion t(3;3)(q21;q26.2) flip GATA2 regulatory element causing EVI1 ectopic activation and GATA2 deregulation, simultaneously [51]. Other chromosomal rearrangements include intrachromosomal deletions of genomic DNA between active enhancers and a proto-oncogene [52,53] and insulator duplication or deletions that are responsible for novel topological and functional chromatin interactions [54].

Most of these studies demonstrate the implication of active SE regions rather than the function of the actual enhancer transcripts. However, there also direct evidence pointing towards eRNAs. Examples are focal somatic CNAs affecting at least one eRNA locus but not protein-coding genes [55] and upregulated eRNAs correlated with CNA [56].

## 6. eRNAs in Cancer

Actively transcribed enhancers regulate most oncogenes and eRNA levels are associated with dysregulated enhancer activation and gene expression in cancer. For example, in sex-hormone-dependent tumors—such as breast and prostate cancers—key regulator eRNAs are induced by hormone receptors and there are many evidences that some eRNAs are prominent factors in sex hormone-dependent cancer development [9,13,57,58,59]. Hormonal regulation of eRNAs might explain the gender association with incidence also in tumors of non-reproductive organs [60]. It is the case of SMAD7e, an estrogen- associated eRNA that contributes to the initiation and progression of bladder cancer [61]. CRISPR-Cas13a technology, applied in bladder cancer cells to downregulate SMAD7e, suppressed proliferation and migration, promoted apoptosis, decreased invasion in bladder cancer, and also impaired the tumor promoting action of estrogen both in vitro and in vivo [61]. Not just sex hormones can modulate enhancer function. Hoffman et al. identified the oncogenic mechanism associated with DDIT4, a gene often dysregulated in a variety of cancer types, showing its modulation by targeting individual regulatory elements within its hormone-responsive super-enhancer [58]. Dexamethasone specifically triggers glucocorticoid receptor binding at different responsive elements in the enhancer, tightly modulating its eRNA expression and therefore regulating DDIT4 transcription [58]. 

The eRNAs inducing oncogene expression can be considered as potential therapeutic targets, as for instance CCAT1, which directly drives MYC expression in colorectal cancer [62] or other eRNAs specifically transcribed at MYC super-enhancer in hepatocellular carcinoma [63]. In acute myeloid leukemia, dihydroergotamine (DHE)—a drug activator of NR4A nuclear receptors—represses the expression of a selected group of SE-associated leukemic oncogenes including MYC [64]. DHE inhibits MYC SE functional activity by eliminating eRNA transcription and enhancer–promoter looping and suppresses tumor growth both in vitro and in vivo [64]. Vice versa, other eRNAs can be considered as tumor suppressors, such as eRNAs induced by p53 mediating p53-dependent gene transcription [65,66]. Some eRNAs downstream the p53 pathway are not directly bound by p53 but by p53-induced lncRNAs [67]. An example is LED, a lncRNA strongly induced by nutlin-3a, activator of p53. LED can in turn activate p21 eRNA transcription at CDKN1A enhancer and mediate the tumor suppressor function of p53 [67].

Nowadays, a systematic mapping of eRNAs expressed in different types of cancer is available thanks to the effort of scientists who tried to infer the cancer-specific expression of eRNAs from the RNAseq data collected in several cancer series world-wide [32,33,45,55,68]. The expression profile of those eRNAs may help in resolving the intra-tumor heterogeneity and improve the diagnosis and treatment of many cancers.

eRNAs are also involved in the development of resistance to anticancer therapies [69]. The cells under treatment can acquire genetic alterations that give them a proliferative advantage, but they can also adapt at an epigenetic level, activating the expression of transcriptional programs that allow cell survival under therapeutic pressure [70,71,72]. Activation of different enhancer networks and eRNA expression are involved in this mechanism [73,74]. eRNAs are clinically relevant because of their cancer-type specific pattern of expression and for such a reason they can represent diagnostic or prognostic markers in cancer therapy [75]. Integrative analysis of multi-omics and pharmacogenomics data across large-scale patient samples and cancer cell lines showed that the majority of genes in the canonical cancer signaling pathways are highly correlated with specific eRNAs in at least one cancer type [76]. The relationship between eRNAs and target genes are confirmed by evidence of chromatin interaction from high-throughput chromosome conformation capture (Hi-C) data. Accordingly, associations between eRNAs and response to anticancer drugs, targeting the linked signaling pathway, emerge crossing data of eRNA expression in 1000 cancer cell lines from Cancer Cell Line Encyclopedia (CCLE), and drug sensitivity available from the Cancer Therapeutics Response Portal (CTRP). A validated example derived from this computational pipeline is NET1e, which is highly expressed in breast cancer and has oncogenic effects in vitro. NET1e in situ overexpression induced drug resistance to BEZ235, a dual PI3K/mTOR inhibitor and obatoclax, pan-BCL2 inhibitor, in breast cancer cell line [76]. Recently, the same research group further improved this approach, estimating the correlation between eRNAs expression with genetic variants, drug response, and immune infiltration in patients to facilitate the functional and clinical investigations of eRNAs in human cancers [45].

## 7. Cold Tumors and the Potential Role of eRNAs 

Enhancer activation and eRNA transcription are also involved in inflammatory response, placing the eRNAs at the interplay between cancer and immune cells. IL1b-eRNA attenuates transcription and release of the pro-inflammatory mediators to influence human innate immune response [77] and CCL2-eRNA regulates the inflammatory macrophage activation [78]. In general, some eRNAs are specifically expressed during activation of lymphocytes [32]. 

Some tumors are defined as cold tumors because of their capacity in evading a strong immune response by a tumor microenvironment (TME) lacking the infiltration of T cells that can kill the tumor cells [79]. Notably, most cancers of the brain, breast, ovary, pancreas, and prostate are considered cold tumors, rarely responding to currently available immunotherapies [80,81]. 

Immune checkpoints are stimulatory or inhibitory regulatory signals that play crucial roles in balancing the activation of adaptive immunity and simultaneously ensuring self-tolerance from autoantigens to prevent autoimmunity [82]. Important stimulatory checkpoint molecules include CD27, CD28, and CD40. Inhibitory checkpoint molecules include, among others, members of the B7-family with the most prominent representative PD-1. To evade immune-cell recognition, some cancer cells downregulate the expression of specific tumor antigens and instead express ligands, as PD-L1, that induce immune cell exhaustion by interacting with PD-1 expressed on immune cells [81].

A growing body of evidence is emerging sustaining the role played by eRNAs in immune escape of tumor cells [83]. A model described by Xiang et al. suggests that the SE-derived eRNA CCAT1 regulates the expression of PD-L1 in tumor cells in cis and trans. First, CCAT1—transcribed from the MYC-515 locus—induces the expression of PD-L1 in consequence of MYC expression regulation by classical in cis enhancer promoter loop [84]. Second, CCAT1 can interact with two transcription factors, TP63 and SOX2, to promote in trans the transcription of EGFR, which induces PD-L1 expression by activating PI3K/AKT and RAS/MAPK pathways [85].

The connection between eRNAs and immune escape has been proven also at clinical levels in more than one cancer type. In a recent paper, the predictive potential of an eRNA, LINC02257, in colon cancer and other tumors was sustained by its correlation with tumor mutational burden (TMB) and with the infiltration of immune cells in the TME [86]. Another large study conducted on 835 prostate cancer specimens showed that eRNAs influence the tumor development via regulating immune processes [87]. Here, the authors constructed a robust prognostic indicator based on expression of three eRNAs and related target genes which promote tumor progression by accelerating immunodepletion [87]. The high and low risk patients, stratified on the basis of this eRNA prognostic indicator, showed different extent of immune cell infiltration into the tumor: the high-risk patients had a lower level of CD8+ cells and the low risk an higher infiltration of CD4+ [87]. Different TMB and microsatellite instabilities were also present in high- and low-risk patients [87]. In invasive breast cancer, the eRNA WAKMAR2 targets genes are involved in several crucial pathways, including cytokine activity, MHC class II protein complexes, and immunoglobulin-mediated immune responses. WAKMAR2 has a positive prognostic value in invasive breast cancer patients, likely due to the relatively high expression of its immune target genes. Moreover, similarly to the above-mentioned LINC02257, WAKMAR2 is correlated with high TMB and, in consequence of that, with the efficacy of immune checkpoint inhibitors [88]. An immune-related eRNA–mRNA axis was identified in lung adenocarcinoma using data obtained from The Cancer Genome Atlas (TCGA), ImmLnc, and ImmPort databases [89]. Patients with a high LINC00987/A2M axis expression have a better prognosis, with an increased proportion of antitumor immune cell infiltration [89] (Figure 2).

## 8. New Insights in cis Transcriptional Regulation Mechanism

The stabilization of enhancer–promoter looping is the most described mechanisms of eRNA action [1,90,91]. The eRNAs taking part in the loop are usually transcribed in sense with the target mRNA. Even though enhancers are generally bidirectionally transcribed, the function of antisense eRNAs is much less clear. In prostate cancer, AS-eRNAs that are expressed at enhancers under the regulation of androgen receptor can recruit DNMT1 at 3′prime of AR-related target genes [92]. As a consequence, antisense transcription at the gene ending is suppressed, enhancing mRNA expression [92]. Notably, the chromatin establishes a double-loop, which enables spatial targeting of sense-eRNA to the promoter and AS-eRNA to gene-end in cis. The C-rich region is the key domain for AS-eRNA function and its binding to DNMT1 [92].

Bai et al. have recently provided pivotal data on the role of sequence similarities in the formation of classical enhancer–promoter loops [93]. Data obtained in different cancer cell lines show that Alu-derived sequences dominate sequence similarities between enhancer and promoter regions [93]. Alu elements are the most abundant transposable elements in the genome, and they are associated with chromatin organization and found in most enhancers [94]. The enhancer–promoter interaction, in presence of similar Alu sequences, is sustained by eRNAs directly interacting with their targeted promoters by structures, called *trans*-acting R-loops [93]. 

R-loops are special three-stranded nucleic acid structures that comprise a nascent RNA hybridized with the DNA template strand, leaving a non-template DNA single-stranded [95]. They often form in GC-rich regions, in particular at the 5′ region of genes, where they are correlated to non-methylated promoters [96], but they are also associated with the regulation of transcription mediated by noncoding antisense RNA [97]. eRNAs are often associated with R-loops [98]. The formation of this kind of structure was described in detail for the eRNA PEARL, which regulates the expression of Pcdhα, one of the clustered protocadherin (Pcdh) genes. Pcdhα, β, and γ, are organized into three closely linked clusters; α and γ have variable and constant genomic organization, like those of the immunoglobulin genes. Those genes are stochastically expressed in a cell-specific manner in the brain, where they act as neural identity code. The eRNA PEARL promotes long-distance chromatin interactions between distal enhancers and target promoters. The local R-loop formation associated with PEARL transcription might stall the “loop extrusion” of the CTCF/cohesin complex, thus bringing the distal enhancer in close contact with variable promoters [99].

Notably, R-loops are associated with genomic instability and DNA damage, suggesting an implication in cancer initiation and progression [100]. A protective mechanism might be the association of R-loops and the RNA exosome, mainly responsible of the eRNAs short half-life. Indeed, studies showed that the genetic depletion of specific subunits of the RNA exosome, here exosome component 3 (Exosc3) and Exosc10 lead to the upregulation of eRNAs, increased R-loop formation, and genomic instability at the enhancers of mouse embryonic stem cells (mESCs) and B cells [101]. These additional observations strengthen the role played by eRNAs in enhancer–promoter-loop formation and the importance of tight regulation of this interplay.

What has been known for a long time is that, upon enhancer activation, the master transcription factors recruit CBP/p300 and the RNApol II leading to the main features of active enhancer: hyperacetylation and eRNA transcription [102]. In the last few years, it has emerged that, besides stabilizing the enhancer–promoter loop, eRNAs can also enhance the activation of the regulatory region. In fact, eRNAs can bind to CBP in a locus-specific manner and stimulate its acetyltransferase activity increasing the affinity for histone substrate, in cis [103]. The maintenance of enhancer activation is further mediated by eRNAs through the direct interaction with BRD4, augmenting its capability to bind to acetylated histones and promote RNApol II elongation, with consequent increase in eRNA transcription itself. This feedforward loop mechanism was well demonstrated in the response to chronic immune signaling in colon cancer cells and resulting in the enhanced transcription of pro-inflammatory genes [104]. In leukemia, SEELA—an eRNA activated by the MLL fusion oncoprotein—promotes the enhancer activity in cancer through its direct binding to the histone H4 [105]. The H4 binding of the eRNA controls its chromatin retention and influences in cis the histone modification reading via BRD4 recruitment, which is impaired by the knock down of the eRNA [105]. New interactions among eRNAs and nuclear proteins are continuously emerging: interestingly the interaction between eRNA and Ago1, an RNA interference silencing complex (RISC) component that is present in mammalian nuclei, has been demonstrated in a muscle cell model [106]. Ago1 controls gene expression and myogenic differentiation by the eRNA-mediated interaction with CBP and consequent regulation of H3K27ac [106]. Ago1 directly interacts with eRNAs, but it also mediates the recruitment of RNApol II to activated enhancers. Thus, in its absence, eRNA transcription itself is reduced [106]. In the same cellular model, a previous study had already added another piece of information [34]. The eRNA seRNA-1, induced by the transcription factor MyoD, specifically binds to the RNA binding protein, hnRNPL protein, and the disruption of such interaction attenuates RNApol II and H3K36me3 deposition at the target genes [34]. CLIP-seq experiments allowed generalizing the mechanism transcriptome-wide in several cell types [34]. The authors also identified the consensus sequence needed for the binding of both eRNA and hnRNLP, implying not a promiscuous mechanism, but an accurate local regulation in gene transcription [34] (Figure 3, Table 1).

## 9. Stabilization of Canonical eRNAs

The majority of eRNAs is expressed in few copies per cell, they are non-polyadenylated, shorter, and less spliced than mRNAs, even if they are both transcribed by RNApol II [107]. This is due to less productivity of RNApol II elongation at eRNA loci. Moreover, the low density of splicing and polyadenylation signals make those transcripts more prone to early termination and RNA exosome degradation [107]. Nevertheless, variations in these features introduce physical and functional heterogeneity in the eRNA class. Enhancers more active than others, with higher level of H3K27ac, are more accessible and easier to be elongated and spliced [107]. This implies that eRNAs, residing on the chromatin to support enhancer–promoter looping with a quick turn over under specific conditions, can acquire more stability and diffuse in the nucleus or even in the cytoplasm. This is what happens to GECPAR, an eRNA associated with POU2AF1 locus and specifically expressed in germinal center (GC) B cells, in which the neighbor protein coding gene plays a pivotal role in transcriptional regulation [108]. By RNA subcellular fractionation, we proved this eRNA is mainly chromatin associated but, when expressed in sufficient copies, is more elongated and polyadenylated and it detaches from the native super enhancer to reach distant sites of action [108]. By capture hybridization analysis of RNA targets (CHART)-seq experiments, we identified the direct targets of GECPAR and its role played in the cancer counterpart of GC B cells, the germinal center diffuse large B cell lymphoma (GCB-DLBCL). GECPAR is anti-correlated to cell proliferation, by in trans transcriptional regulation of several genes involved in cell growth and differentiation [108]. Finally, we propose GECPAR as a key surveillant of the GC maintenance: induced by BCR activation, it retains B cells in the GC light zone, reducing the tendency to re-enter in the dark zone, blocking MYC, or to exit and differentiate to plasma cells, repressing PRDM1 [108]. GECPAR also reduces B-cell proliferation rate by directly inducing TLE4, a negative repressor of LEF1, mediator of Wnt pathway, blocking the cross talk between Wnt and NFkB [108]. This is an example of an eRNA transcribed from a strongly active super enhancer, which acquires capability of trans regulation, but it is not the only lncRNA which acts at distal loci: other well-known examples are HOTAIR [109], Paupar [110], and Bloodlinc [110,111] (Table 1).

## 10. eRNA Activity beyond Promoting Gene Transcription 

Recent works provide novel insights into different functions of eRNAs and suggest that eRNAs not only play a role at the pre-transcriptional level, but they can also function at the post-transcriptional level.

An example reported is the identification of a novel eRNA, named NEAT1-MALAT1-Locus enhancer RNA (eNEMAL) [112]. eNEMAL is transcribed from the MALAT1 enhancer locus, between MALAT1 and NEAT1 loci, and it is upregulated in response to hypoxia in various breast cancer cell lines [112]. Variation in eNEMAL level leads to minor changes in MALAT1 expression, but a significantly higher abundance of the long isoform of NEAT1-lncRNA (NEAT1_2) is seen when eNEMAL is upregulated [112]. NEAT1_2 isoform is a critical component of paraspeckles, a nuclear sub-compartment involved in gene regulation and important for cellular survival [112]. Importantly, paraspeckles are enriched in breast cancer cells under hypoxia [112]. eNEMAL promotes NEAT1_2 generation, potentially by inhibiting the polyadenylation process required for the production of NEAT1_1 [112]. Alternative polyadenylation of NEAT1 is responsible for the switch between NEAT1_1 and NEAT1_2 and it is regulated by several factors, as CPSF6 and NUTD21 that promote polyadenylation and hnRNPK that blocks it [112]. eNEMAL is believed to play a role in this regulatory mechanism [112]. 

More evidence linking eRNA expression and regulation of splicing was provided by the characterization of the eRNA transcribed from the enhancer located 157 kb downstream VEGFA [113]. This gene is involved in many solid tumors since it promotes new vascularization of tumors, but it is also important for the growth of hematological malignancies. Its enhancer is active in embryogenesis and then switched off by methylation, but in chronic myeloid leukemia, it is hypomethylated and, consequently, VEGFA overexpressed. VEGF is present in different isoforms and their alternative splicing is often deregulated in cancer cells [113]. The alternative splicing is correlated with the processivity of RNApol II and in chronic myeloid leukemia this mechanism is altered by the expression of the eRNA +157 [113]. Reduction in enhancer activity impairs RNApol II elongation and this favors the exclusion of exon 6 and exon 7, giving rise to the isoform VEGFA121. On the other hand, the activation of the enhancer increases RNApol II elongation rate and promotes the exon 6–7 retention, causing the expression of VEGFA189. The protein CCNT2 also plays a role as chromatin binding factor promoting RNApol II elongation in response to the interaction between the VEGFA promoter and the +157 enhancer. An additional layer of relevance of this research is that high expression of +157 eRNA and inclusion of VEGFA exons 6a and 7 detected in peripheral blood of CML patients with high levels of VEGFA in the plasma. Both eRNA and inclusion of exons 6a and 7 are reduced in patients in remission, linking the +157 enhancer activity with the change of alternative splicing [113].

A similar mechanism linking eRNAs and complex molecular processes is associated with the developmentally controlled DNA rearrangement of the immunoglobulin (Ig) genes in mammals [114]. A specific eRNA called LncRNA-CSRIgA promotes topological chromatin changes in Ig loci to favor class-switch recombination (CSR) in GC B-cells. LncRNA-CSRIgA facilitates the recruitment of the molecular machinery near CTCF sites, which are important for class switch loop formation, and promotes interaction between the locus sites to finally induce an IgA switch [114]. 

eRNAs present some secondary structural properties that promote macromolecular complex formation. Investigating eRNA transcription from a structural perspective should help in identifying structural RNA elements that are involved in diverse cellular processes and that could lead to the design of targeted agents. Ren et al. [33] first identified 23,878 eRNA regions across 50 human cell and tissue types using genome-wide chromatin immunoprecipitation-sequencing (ChIP-seq) and RNA-seq data of the Roadmap Epigenomics Project [115]. Then, focusing on eRNA regions defined in 10 normal lymphoid primary cells, they reported 18,767 unique structural ncRNAs with known secondary structure [33]. Autoimmune disease-associated single nucleotide polymorphisms were also analyzed to predict their impact on local RNA secondary structure, showing 1764 structural ncRNAs in lymphoid enhancers as riboSNitches, RNA with large structural disparities caused by single nucleotide variants. This evidence strengthens the role of eRNA structures to the execution of a specific function, the lack of which may be associated with a pathological condition [33]. (Table 1)

## 11. eRNAs within the Cytoplasm Promoting Cancer Progression and Therapy Resistance 

Urothelial Cancer Associated 1 (UCA1) is a super-enhancer derived eRNA expressed in the early embryo development, but also expressed in cancer [116,117]. Notably, UCA1 is a polyadenylated eRNA acting in the cytoplasm [116]. UCA1 overexpression drives ovarian cancer development by activating the Hippo–YAP pathway [116]. It enhances the interaction between YAP and the phosphorylated form of its positive regulator, AMOTp130, that undergoes a conformational change induced by UCA1 binding, which outcompetes pLATS1/2 for YAP binding [116]. The pLASTS1/2-YAP complex formation leads to phosphorylation of YAP and cytoplasmic retention, while AMOTp130-UCA1 binding leads to increased dephosphorylation of YAP, promoting nuclear translocation [116]. In the nucleus, YAP acts as a transcription factor and accelerates expression of the respective oncogenic target genes [116]. This study identified a protein–eRNA interaction and suggested a linker function of UCA1 for advantageous protein-complex formation. Additionally, UCA1 is associated with other pathways that can promote cancer pathogenesis and therapy resistance, such as cisplatin resistance in ovarian cancer and additional solid tumors [116,117]. In gastric cancer cells, UCA1-mediated cisplatin resistance appears as consequence of EZH2 recruitment and increased activation of the PI3K/AKT pathway that counteracts the cisplatin-induced apoptosis induction [117]. Hence, treatments targeting UCA1 might provide potentially new therapeutic strategies for cisplatin resistance in multiple tumor types [116,117].

As seen for UCA1, eRNAs represent an intriguing class of molecules interacting with different partners, cell compartments and operating multiple mechanisms of action. An example is given by FAL1, an oncogenic eRNA overexpressed in various cancers, in which it supports cell proliferation, migration, and invasion [118]. The oncogenic effects of FAL1 are multiple. In non-small-cell lung carcinoma and ovarian cancer, FAL1 stabilizes BMI1, a core protein of the Polycomb repressive complex 1 (PRC1), which facilitates epithelial–mesenchymal transition by modulating the PTEN/AKT pathway [118]. In colorectal cancer, FAL1 promotes cancer growth and metastasis via STAT3 phosphorylation [118]. Additionally, FAL1 can act as a competing endogenous RNA (ceRNA) binding miR-637, a tumor suppressor miRNA that negatively regulates the oncogenic nuclear protein 1 (NUPR1) expression in colorectal cancer cells [118]. In this scenario, FAL1 upregulation would suppress miR-637, with upregulation of NUPR1 that then increases the expression of hypoxia inducible factor-1 alpha (HIF-1α) [118]. Finally, in osteosarcoma, FAL1 contributes to the growth and metastatic potential of cancer cells by inducing EMT-markers and phosphorylation of GSK-3β, a protein crucial in Wnt signaling pathway regulation [118]. Table 1.

## 12. Conclusions and Perspectives 

eRNAs are emerging as a very heterogeneous group of RNA molecules due to a variety of molecular mechanisms of action. Their expression is tightly regulated and strongly tissue specific, in concordance with the pivotal role they play in the maintenance of the cell state. The short half-life and the chromatin association are the common features that make eRNAs difficult to characterize in the first place, but meanwhile determine the precise spatial–temporal localization so important for their function. Nowadays, eRNAs can also be detected in primary specimens and they potentially allow us to monitor the evolution of the disease during treatment exposure or cancer progression. They can indicate new oncogenic pathways activated in the tumor cells and suggest potential new therapeutic targets for treatment combinations. Understanding the mechanistic contributions of individual regulatory elements to oncogene expression is fundamental to develop prospective therapeutics. Furthermore, eRNA tendency to assume secondary structures makes them perfect switches to regulate the interaction of intermolecular complexes. This feature identifies eRNAs as potential therapeutic targets. The recent evidence of stabilization of some eRNAs that translates into the ability to diffuse in the cell and take part in several complex molecular mechanisms highlights their relevance as entities induced into the cell in response to specific stimuli and that can become actionable targets. The real limitation still to overcome is the insufficient knowledge of reliable structural conformations of eRNAs. Understanding of dynamic changes of RNA structures, in presence of sequence variations or when involved in intermolecular interactions, is needed to design small-molecule modulators or oligonucleotides that may specifically target eRNAs. Therefore, we conclude that the field of eRNAs and enhancer-derived ncRNAs offer considerable scientific and medical potential and deserves further intense investigations with respect to cancer and other diseases.

## Figures and Tables

**Figure 1 cancers-14-01978-f001:**
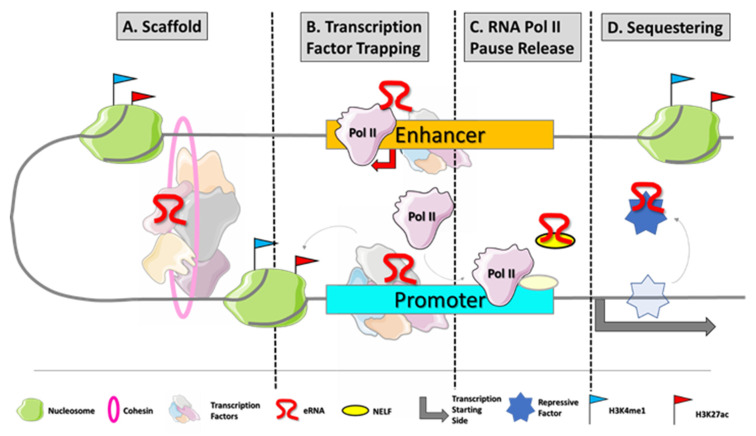
eRNA biogenesis and frequently described modes of action. Enhancers are typically disengaged of nucleosomes which makes their DNA sequences easily accessible for transcription factor (TF) binding. Master TFs occupy enhancer regions at specific DNA motifs to initiate the transcriptional process by recruiting RNApol II at enhancers. Those transcripts mainly promote gene expression by different mechanisms of action: (**A**) eRNAs can interact with transcription factors to support proper enhancer–promoter loop formation [8,9]. (**B**) eRNAs participate in TF trapping, RNApol II loading, and histone modification [10]. (**C**) eRNAs can assist RNApol II pause release by binding negative elongation factor (NELF) [11]. (**D**) eRNAs can act as decoys for repressive co-factors of transcription which would bind at the respective target genes.

**Figure 2 cancers-14-01978-f002:**
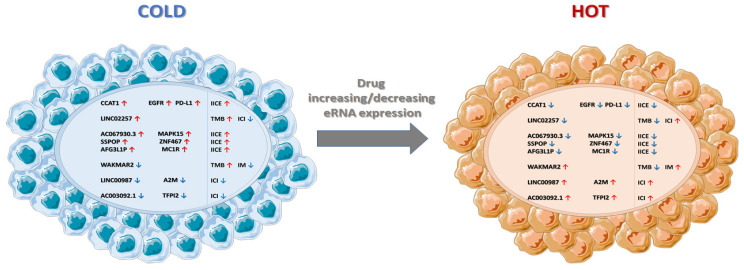
Cold tumors and the potential role of eRNAs. Some tumors are defined as cold tumors due to their capacity in evading a strong immune response with a tumor microenvironment, which suppresses tumor infiltrating T cells. The figure summarizes the growing literature sustaining the role played by eRNAs in immune escape of tumor cells. Increasing or decreasing the expression of specific eRNAs by novel drugs might switch cold tumors to hot tumors and improve current anti-cancer therapies in combined treatments. ICI = immune cell infiltration; IICE = inhibitor immune-checkpoint expression; IM = immuno-modulation; TMB = tumor mutational burden.

**Figure 3 cancers-14-01978-f003:**
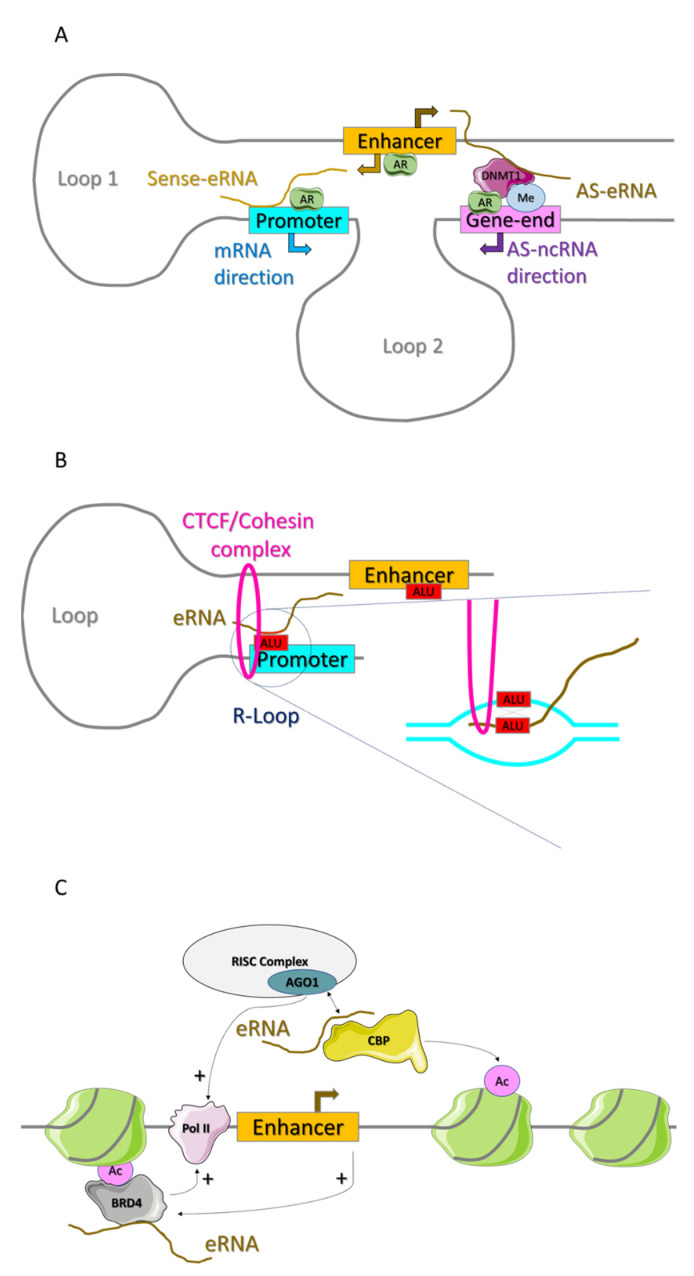
New insights in cis transcriptional regulation mediated by eRNAs. (**A**) In prostate cancer, AS-eRNAs regulated by androgen receptor (AR) can recruit DNMT1 to the 3′ of AR-related target genes, suppressing tail-to-tail antisense transcription and enhancing mRNA expression. Here, the chromatin establishes a double-loop, which enables spatial targeting of sense-eRNA and AS-eRNA to the promoter or gene-end in cis, respectively [92]. (**B**) The enhancer–promoter interaction, in presence of similar Alu sequences, is sustained by eRNAs directly interacting with their targeted promoters by structures, called *trans*-acting R-loops [93]. (**C**) eRNAs contribute to the maintenance of enhancer activation by direct interaction with CBP and BRD4, increasing histone acetylation and consequently BRD4 binding to acetylated histones [103,104]. This feed-forward loop promotes RNApol II elongation and eRNA transcription. Additionally, Ago1 controls gene expression and differentiation by the eRNA-mediated interaction with CBP and consequent regulation of H3K27ac. Ago1 also mediates the recruitment of RNApol II to activated enhancers [106].

**Table 1 cancers-14-01978-t001:** Evidence and novel insights on the role and function of eRNAs.

Specifc eRNAs	Notes on eRNAs	Biological Implications	Mechanism of Action	References
Peblr20 LncRNA	Expression of Peblr20 activates stemness-like genes, such as Pou5F1, in fibroblasts.	Differentiation of pluripotent stem cells	Recruiting of TET2 at modulate enhancer–promoter looping.	[20]
	Chromatin opening follows C/EBPa binding induce cell specific eRNAs activation.	Transdifferentiation of human leukemia B-cells	Synergistic cooperation of different eRNAs contribute to target genes transcription.	[19]
	m6A eRNA, bound by YTHDC1, helps chromatin condensation, cross talking to other coactivators and impacting enhancer activity.	Modulation of transcription processes	Sequence specific mechanisms such as high m6A levels correlate with long and more active enhancers.	[22]
Antisense RNA KHPS1	KHPS1 forms an RNA–DNA triplex at the SPHK1enhancer. Enhancer is required for SPHK1 expression and cell proliferation.	Cell prolifereation and viability	Triplex-based recruitment of chromatin-modifying complexes. Promoter activation involves the eRNA- mediated eviction of CTCF.	[24]
	Enhancer release and retargeting (ERR) is at the basis of disease-causing genes activation.	Activation of disease-causing genes	Binding of CTCF or deletion in promoter determine alternative gene activation.	[27]
CCAT2 eRNA	MYC, miR-17-5p, and miR-20a are up-regulated by CCAT2.	Metastatic progression and SNP-related risk in colon cancer	SNP status affects CCAT2 expression.	[31]
p53 enhancer regions (p53BERs)	eRNA production is required in p53 transcription enhancement.	Tumor suppressor gene regulation	Enhancers interact intrachromosomally with multiple gene.	[32]
IL1b-eRNA	Attenuates transcription and release of the proinflammatory mediators.	Innate immune response	In *cis*/*trans* gene modulation, specific mechanisms still to be elucidated.	[43]
CCL2-eRNA	CCL2-eRNA regulates the inflammatory macrophage activation.	Innate immune response	Enhancer activity linked to inflammatory gene expression via modulating CBP-mediated H3K27acetylation.	[44]
CCAT1 1L lncRNA	Regulate the expression of PD-L1 in tumor cells.	Immune escape in cancer	In *cis*/*trans* promoter/enhancer looping.	[50]
LINC02257	Correlation with tumor mutational burden and infiltration of tumor microenvironment.	Immune escape in colorectal cancer	Unknown	[52]
WAKMAR2	Expression of eRNAs and related target genes promote immunodepletion.	Immune escape and breast cancer	Not specified	[54]
LINC00987	LINC00987/A2M is involved in the proportion of antitumor immune cell infiltration.	Immune escape and prognosis in lung adenocarcinoma	Unknown	[55]
PSA eRNAs	Antisense eRNAs are expressed and functional upon androgen receptor (AR) activation.	Therapy and diagnosis in prostate cancer	eRNAs–DNMT1 interaction in chromatin looping.	[58]
	eRNA and Alu elements associate in enhancer–promoter interactions.	Transcriptional regulation in mammals	Alu sequences sustain *trans*-acting R-loops.	[59]
PEARL eRNA	eRNA PEARL, which regulates the expression of Pcdhα, one of the clustered protocadherin (Pcdh) genes.	Transcriptional regulation in mammals	Promoter/enhancer looping mediated by CTCF/cohesin complex.	[65]
	Exosome component 3 (Exosc3) and Exosc10, lead to the upregulation of eRNAs with increased R-loop formation.	Genomic instability in embryonic stem cells (mESCs) and B cells	Promoter/enhancer looping	[67]
SEELA eRNAs	Chromatin interaction with histone modifiers is mediated by enhancer activity.	Cancer initiation and progression in MLL leukemia	H4 binding and BRD4 recruitment	[71]
	Ago1 is directly interacting with eRNAs but it also mediates the recruitment of RNApol II to activated enhancers.	Myogenic differentiation	eRNA interact with Ago1 and other nuclear proteins	[72]
GECPAR	eRNA is mainly chromatin associated but, when expressed in sufficient copies, is more elongated and polyadenilated, and it detaches from the native super enhancer to reach distant sites of action.	Differentiation and proliferation role of germinal center (GC) B cells	Stabilization of canonical eRNAs with acquire *trans* activity.	[74]
eNEMAL	eNEMAL is transcribed from the MALAT1 enhancer locus, and is upregulated in response to hypoxia.	Post-transcriptional regulation in various breast cancer cell lines	Alternative polyadenylation of NEAT1 produce a specific isoform with a critical role in paraspeckles.	[78]
VEGFA eRNA	Reduction in enhancer activity impairs RNApol II elongation and this favors the exclusion of exon 6 and exon 7, giving rise to the isoform VEGFA.	Post-transcriptional regulation in hematological malignancies	POLII–eRNA interaction regulates alternative splicing.	[79]
LncRNA-CSRIgA	Promotes topological chromatin changes in Ig loci to favor class-switch recombination (CSR) in GC B-cells.	Class-switch recombination (CSR) in GC B-cells	eRNA facilitates the recruitment of the molecular machinery near CTCF sites, which are important for class switch loop formation.	[80]
UCA1 eRNA	UCA1 overexpression drives ovarian cancer (OC) development by activating the Hippo–YAP pathway.	Pathogenesis of ovarian cancer and cis-platin resistance in gastric cancer	AMOTp130–UCA1 interaction leads to increased dephosphorylation of YAP, promoting nuclear translocation and oncogene activation; LncRNA UCA1 promotes cisplatin resistance in gastric cancer via recruiting EZH2 and activating PI3K/AKT pathway.	[82,83]
FAL1 eRNA	FAL1 is an oncogenic eRNA and is overexpressed in various malignancies where supports cell proliferation.	Cell proliferation and metastasis in different malignancies	Stabilization of BMI1 that facilitates EMT by modulating PTEN/AKT pathway.	[84]

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
