# Peer review of "Enhancer RNAs (eRNAs) in Cancer: The Jacks of All Trades"

_cancers, 2022, doi:10.3390/cancers14081978_

Round 1
Reviewer 1 Report
Overall a fine review of a rapidly expanding field.
I would have one minor suggestion that I think would elevate the manuscript as follows:
Please add a section/figure on the area of how chromosomal rearrangements can affect aberrant enhancer activity, in other words "enhancer hijacking".
Author Response
Overall a fine review of a rapidly expanding field.
I would have one minor suggestion that I think would elevate the manuscript as follows:
Please add a section/figure on the area of how chromosomal rearrangements can affect aberrant enhancer activity, in other words "enhancer hijacking".
We thank the reviewer for the positive comment and we followed the suggestion of including a section about "enhancer hijacking". It is the red text in now section 4.
Reviewer 2 Report
Napoli et al. describe the very interesting and novel topic of enhancer RNAs.
The manuscript is well written despite a minimal language review is recommended. E.g. line 60, the authors write "eRNAs seem to play not only a role in homeostasis", while it should be "eRNAs seem to play a role not only in homeostasis".
My main request is that a section describing the molecular features of eRNAs (as of now in section #7) such as polyAdenylation, splicing, capping (typical for RNA pol II transcripts) in addition to the methods used to assess their expression is missing (I would put it as #2 after the introduction).
Lines 42-44: the description of (all?) enhancers as having H3K27ac is not completely accurate. The distinction between active, poised and primed based on chromatin marks has been correctly explained in lines 116-121. You should either specify that you are referring to active enhancers or add the other possible states.
Figure 2, panels A, and C: the yellowish bent arrow representing the TSS of the eRNA suggests that the DNA surrounding the enhancer is transcribed, but not the DNA corresponding to the enhancer itself. Is that what the authors meant?
Author Response
Napoli et al. describe the very interesting and novel topic of enhancer RNAs. The manuscript is well written despite a minimal language review is recommended. E.g. line 60, the authors write "eRNAs seem to play not only a role in homeostasis", while it should be "eRNAs seem to play a role not only in homeostasis".
We thank the reviewer for the positive comment and we followed the recommendation of language review. We revised the text carefully and corrected some errors, including the one at the previous line 60, as referred by the reviewer.
My main request is that a section describing the molecular features of eRNAs (as of now in section #7) such as polyAdenylation, splicing, capping (typical for RNA pol II transcripts) in addition to the methods used to assess their expression is missing (I would put it as #2 after the introduction).
We agree with the reviewer's request. We added a section ( section 2, highlighted in red) fully dedicated to molecular features of eRNAs and methods for their detection.
Lines 42-44: the description of (all?) enhancers as having H3K27ac is not completely accurate. The distinction between active, poised and primed based on chromatin marks has been correctly explained in lines 116-121. You should either specify that you are referring to active enhancers or add the other possible states.
The reviewer is right and we apology for the imprecision in the previous text. Now we corrected the text and specified we were referring to active enhancers.
Figure 2, panels A, and C: the yellowish bent arrow representing the TSS of the eRNA suggests that the DNA surrounding the enhancer is transcribed, but not the DNA corresponding to the enhancer itself. Is that what the authors meant?
The reviewer is right, the previous figure was misleading. We corrected the figure accordingly.
Reviewer 3 Report
In the present review entitled “Enhancer RNAs (eRNAs) in cancer: the jacks of all trades” the authors have comprehensively reported molecular mechanisms linked to eRNAs activity. This is an interesting review article, describing major achievements in this emerging field.
The authors should address the following minor issues in order to be able to publish their review article.
- The authors should include a figure illustrating eRNA biogenesis and most evidenced modes of action.
- The authors are suggested to include more information on the interesting field of “4. eRNAs in cancer”, in terms of role in oncogenesis and clinical relevance.
- The authors should include a table displaying the reported eRNAs in different human malignancies
- The authors should discuss future perspectives in this field.
Author Response
In the present review entitled “Enhancer RNAs (eRNAs) in cancer: the jacks of all trades” the authors have comprehensively reported molecular mechanisms linked to eRNAs activity. This is an interesting review article, describing major achievements in this emerging field.
We thank the reviewer for the positive comment.
The authors should address the following minor issues in order to be able to publish their review article.
- The authors should include a figure illustrating eRNA biogenesis and most evidenced modes of action. We thank the reviewer for the suggestion. We included the mentioned figure, which is now Figure 1.
- The authors are suggested to include more information on the interesting field of “4. eRNAs in cancer”, in terms of role in oncogenesis and clinical relevance. We thank the reviewer for the suggestion and we furthered extended the section 5, "eRNAs in cancer". The updated text is highlighted in red.
- The authors should include a table displaying the reported eRNAs in different human malignancies. We agree with the reviewer about the interest regarding the roles played by eRNAs in many different human malignancies. Anyway, we believe that an additional table, describing all the reported diseases linked to an eRNA, requires also a section in the text for further explanation . This could be a little far away from our purpose to focus on the role of eRNAs in cancer.
- The authors should discuss future perspectives in this field. We agreed with the reviewer and we extended the last section including also the perspective in the field.
Round 2
Reviewer 2 Report
The authors answered all the reviewer's requests.